# Be Your Own Neighborhood: Detecting Adversarial Example by the Neighborhood Relations Built on Self-Supervised Learning

## Abstract

Deep Neural Networks (DNNs) have achieved excellent performance in various fields. However, DNNs' vulnerability to Adversarial Examples (AE) hinders their deployments to safety-critical applications. This paper presents a novel AE detection framework, named **Beyond**, for trustworthy predictions. **Beyond** performs the detection by distinguishing the AE's abnormal relation with its augmented versions, i.e. neighbors, from two prospects: representation similarity and label consistency. An off-the-shelf Self-Supervised Learning (SSL) model is used to extract the representation and predict the label for its highly informative representation capacity compared to supervised learning models. For clean samples, their representations and predictions are closely consistent with their neighbors, whereas those of AEs differ greatly. Furthermore, we explain this observation and show that by leveraging this discrepancy **Beyond** can effectively detect AEs. We develop a rigorous justification for the effectiveness of **Beyond**. Furthermore, as a plug-and-play model, **Beyond** can easily cooperate with the Adversarial Trained Classifier (ATC), achieving the state-of-the-art (SOTA) robustness accuracy. Experimental results show that **Beyond** outperforms baselines by a large margin, especially under adaptive attacks. Empowered by the robust relationship built on SSL, we found that **Beyond** outperforms baselines in terms of both detection ability and speed.

## 1 Introduction

Deep Neural Networks (DNNs) have been widely adopted in many fields due to their superior performance. However, DNNs are vulnerable to Adversarial Examples (AEs), which can easily fool DNNs by adding some imperceptible adversarial perturbations. This vulnerability prevents DNN from being deployed in safety-critical applications such as autonomous driving (Cococcioni et al., 2020) and disease diagnosis (Kaissis et al., 2020), where incorrect predictions can lead to catastrophic economic and even loss of life.

Existing defensive countermeasures can be roughly categorized as adversarial training, input purification (Mao et al., 2021), and AE detection (Xu et al., 2017). Adversarial training is known as the most effective defense technique (Croce & Hein, 2020), but it brings about degradation of accuracy and additional training cost, which are unacceptable in some application scenarios. In contrast, input transformation techniques are without high training/deployment costs, but their defensive ability is limited, i.e. easily defeated by adaptive attacks (Croce & Hein, 2020).

Recently, a large number of AE detection methods have been proposed (Zuo & Zeng, 2021). Some methods detect AE by interrogating the abnormal relationship between AE and other samples. For example, Deep k-Nearest Neighbors (DkNN) (Papernot & McDaniel, 2018) compares the DNN-extracted features of the input image with those of its k nearest neighbors layer by layer to identify AE, leading to a high inference cost. Instead of comparing all features, Latent Neighborhood Graph (LNG) (Abusnaina et al., 2021) employs a Graph Neural Network to make the comparison on a neighborhood graph, whose nodes are pre-stored embeddings of AEs together with those of the clean ones extracted by DNN, and the edges are built according to distances between the input node and every reference node.

Figure 1: Pipeline of the proposed **BEYOND** framework. First, we augment the input image to obtain a bunch of its neighbors. Then, we perform the label consistency detection mechanism on the classifier's prediction of the input image and that of neighbors predicted by SSL's classification head. Meanwhile, the representation similarity mechanism employs *cosine distance* to measure the similarity among the input image and its neighbors (left). The input image with poor label consistency or representation similarity is flagged as AE (right).

Though more efficient than DkNN, LNG suffers from some weaknesses: some AEs are required to build the graph, so its detection performance relies on the reference AEs and cannot effectively generalize to unseen attacks. More importantly, both DkNN and LNG can be bypassed by adaptive attacks, in which the adversary has full knowledge of the detection strategy.

We observe that one cause for adversarial vulnerability is the lack of feature invariance (Jiang et al., 2020), i.e., small perturbations may lead to undesired large changes in features or even predicted labels. On the other hand, Self-Supervised Learning (SSL) (Chen & He, 2021) models learn data representation consistency under different data augmentations, which intuitively can mitigate the issue of lacking feature invariance and thus improve adversarial robustness.

As an illustration, we visualize the SSL-extracted representation of the clean sample, AE, and that of their corresponding augmentations in Fig. 1 (right). We can observe that the clean sample has closer ties with its neighbors, reflected by the higher label consistency and representation similarity. However, the AE representation stays quite far away from its neighbors, and there is a wide divergence in the predicted labels.

Inspired by the above observations, we propose a novel AE detection framework, named **BE Your Own NeighborhooD** (**BEYOND**). The contributions of this work are summarized as follows:

- We propose BEYOND, a novel AE detection framework, which takes advantage of an SSL model's robust representation capacity to identify AE by referring to its neighbors. To our best knowledge, BEYOND is the first work that leverages an SSL model for AE detection without prior knowledge of adversarial attacks or AEs.

- We develop a rigorous justification for the effectiveness of BEYOND, and we derive an indicator to evaluate the validity of the candidate augmentation.

- BEYOND can defend effectively against adaptive attacks. To defeat the two detection mechanisms: label consistency and representation similarity simultaneously, attackers have to optimize two objectives with contradictory directions, resulting in gradients canceling each other out.

- As a plug-and-play method, BEYOND can be applied directly to any image classifier without compromising accuracy or additional retraining costs.

Experimental results show that BEYOND outperforms baselines by a large margin, especially under adaptive attacks. Empowered by the robust relation net built on SSL, we found BEYOND outperforms baselines in terms of both detection ability and implementation costs.

## 2 BEYOND: PROPOSED METHOD

### 2.1 METHOD OVERVIEW

**Components.** BEYOND consists of three components: a SSL feature extractor $f(\cdot)$, a classification head $g(\cdot)$, and a representation head $h(\cdot)$, as shown in Fig. 1 (left). Specifically, the SSL feature

extractor is a Convolutional Neural Network (CNN), pre-trained by specially designed loss, e.g. contrastive loss, without supervision[1]. A Fully-Connected layer (FC) acts as the classification head $g(\cdot)$, trained by freezing the $f(\cdot)$. The $g(\cdot)$ performs on the input image's neighbors for label consistency detection. The representation head $h(\cdot)$ consisting of three FCs, encodes the output of $f(\cdot)$ to an embedding, i.e. representation. We operate the representation similarity detection between the input image and its neighbors.

**Core idea.** Our approach relies on robust relationships between the input and its neighbors for the detection of AE. The key idea is that adversaries may easily attack one sample's representation to another submanifold, but it is difficult to totally shift that of all its neighbors. We employ the SSL model to capture such relationships since it is trained to project input and its augmentations (neighbors) to the same submanifold (Chen & He, 2021).

**Selection of neighbor number.** Obviously, the larger the number of neighbors, the more stable the relationship between them, but this may increase the overhead. We choose 50 neighbors for BEYOND, since larger neighbors no longer significantly enhance performance, as shown in Fig. 4.

**Workflow.** Fig. 1 shows the workflow of the proposed BEYOND. When input comes, we first transform it into 50 augmentations, i.e. 50 neighbors. Note that BEYOND is not based on random data augmentation. Next, the input along with its 50 neighbors are fed to SSL feature extractor $f(\cdot)$ and then the classification head $g(\cdot)$ and the representation head $h(\cdot)$, respectively. For the classification branch, $g(\cdot)$ outputs the predicted label for 50 neighbors. Later, the label consistency detection algorithm calculates the consistency level between the input label (predicted by the classifier) and 50 neighbor labels. When it comes to the representation branch, the 51 generated representations are sent to the representation similarity detection algorithm for AE detection. If the consistency of the label of a sample or its representation similarity is lower than a threshold, BEYOND shall flag it AE.

---

**Algorithm 1** BEYOND detection algorithm

---

**Input**: Input image $x$, target classifier $c(\cdot)$, SSL feature extractor $f(x)$, classification head $g(x)$, projector head $h(x)$, label consistency threshold $\mathcal{T}_{label}$, representation similarity threshold $\mathcal{T}_{rep}$, Augmentation $Aug$, neighbor indicator $i$, total neighbor $k$
**Output**: `reject / accept`

1: ***Stage1: Collect labels and representations.***
2: $\ell_{cls}(x) = c(x)$
3: **for** $i$ in $k$ **do**
4:     $\hat{x}_i = Aug(x)$
5:     $\ell_{ssl}(\hat{x}_i) = f(g(\hat{x}_i)); r(x) = f(h(x)); r(\hat{x}_i) = f(h(\hat{x}_i))$
6: ***Stage2: Label consistency detection mechanism.***
7: **for** $i$ in $k$ **do**
8:     **if** $\ell(\hat{x}_i) == \ell(x)$ **then** $\text{Ind}_{\text{label}} += 1$
9: ***Stage3: Representation similarity detection mechanism.***
10: **for** $i$ in $k$ **do**
11: **if** $cos(r(x), r(\hat{x}_i)) < \mathcal{T}_{cos}$ **then** $\text{Ind}_{rep} += 1$
12: ***Stage4: AE detection.***
13: **if** $\text{Ind}_{label} < \mathcal{T}_{label}$ or $\text{Ind}_{rep} < \mathcal{T}_{rep}$ **then** `reject`
14: **else** `accept`

---

## 2.2 DETECTION ALGORITHMS

For enhanced AE detection capability, BEYOND adopted two detection mechanisms: *Label Consistency*, and *Representation Similarity*. The detection performance of the two combined can exceed any of the individuals. More importantly, their contradictory optimization directions hinder adaptive attacks to bypass both of them simultaneously.

**Label Consistency.** We compare the classifier prediction, $\ell_{cls}(x)$, on the input image, $x$, with the predictions of the SSL classification head, $\ell_{ssl}(\hat{x}_i), i = 1 \ldots k$, where $\hat{x}_i$ denotes the $i$th neighbor, $k$ is the total number of neighbors. If $\ell_{cls}(x)$ equals $\ell_{ssl_i}(\hat{x}_i)$, the label consistency increases by one, $\text{Ind}_{\text{Label}} += 1$. Once the final label consistency is less than the threshold, $\text{Ind}_{\text{Label}} < \mathcal{T}_{label}$, the *Label Consistency* flags it as AE. We summarize the label consistency detection mechanism in Algorithm. 1.

---

[1]Here, we employ the SimSiam (Chen & He, 2021) as the SSL feature-extractor for its decent performance.

**Representation Similarity.** We employ the *cosine distance* as a metric to calculate the similarity between the representation of input sample $r(x)$ and that of its neighbors, $r(\hat{x}_i), i = 1, ..., k$. Once the similarity, $cos(r(x), r(\hat{x}_i))$, is smaller than a certain value, representation similarity increases by 1, $\text{Ind}_{\text{Rep}}+ = 1$. If the final representation similarity is less than a threshold, $\text{Ind}_{\text{ReP}} < \mathcal{T}_{rep}$, the *representation similarity* flag the sample as an AE. Algorithm. 1 concludes the representation similarity detection mechanism.

Note that, we select the thresholds, i.e. $\mathcal{T}_{label}, \mathcal{T}_{rep}$, by fixing the False Positive Rate (FPR)@5%, which can be determined only by clean samples, and the implementation of our method needs no prior knowledge about AE.

## 3 THEORETICAL JUSTIFICATION

### 3.1 THEORETICAL ANALYSIS

Given a clean sample $x$, we receive its feature $f(x)$ lying in the feature space spanned by the SSL model. We assume that benign perturbation, i.e. random noise, $\hat{\delta}$, with bounded budgets causes minor variation, $\hat{\varepsilon}$, on the feature space, as described in Eq. 1:

$$f(x + \hat{\delta}) = f(x) + \nabla f(x)\hat{\delta} = f(x) + \hat{\varepsilon}, \tag{1}$$

where $\|\hat{\varepsilon}\|_2$ is constrained to be within a radius $r$. In contrast, when it comes to AE, $x_{adv}$, the adversarial perturbation, $\delta$, can cause considerable change, due to its maliciousness, that is, it causes misclassification and transferability (Demontis et al., 2019; Liu et al., 2021; Papernot et al., 2016), as formulated in Eq. 2.

$$f(x_{adv}) = f(x + \delta) = f(x) + \nabla f(x)\delta = f(x) + \varepsilon, \tag{2}$$

where $\|\varepsilon\|_2$ is significantly larger than $\|\hat{\varepsilon}\|_2$ formally, $\lim_{\hat{\varepsilon} \to 0} \frac{\varepsilon}{\hat{\varepsilon}} = \infty$. SSL model is trained to generate close representations between an input $x$ and its augmentation $x_{aug} = Wx$ (Hendrycks et al., 2019; Jaiswal et al., 2020), where $W \in \mathbb{R}^{w \times h}$, $w$, $h$ denote the width and height of $x$, respectively. Based on this natural property of SSL ($f(Wx) \approx f(x)$), we have:

$$f(Wx) = f(x) + o(\hat{\varepsilon}), \nabla f(Wx) = \nabla f(x) + o(\hat{\varepsilon}), \tag{3}$$

where $o(\hat{\varepsilon})$ is a high-order infinitesimal item of $\hat{\varepsilon}$. Moreover, according to Eq. 1 and Eq. 3, we can derive that:

$$\begin{aligned} f(W(x + \hat{\delta})) &= f(Wx) + \nabla f(Wx)W\hat{\delta} \\ &= f(x) + \nabla f(x)W\hat{\delta} + o(\hat{\varepsilon}). \end{aligned} \tag{4}$$

We let $\hat{\varepsilon}_{aug} = \nabla f(x)W\hat{\delta}$ and assume $\hat{\varepsilon}_{aug}$ and $\hat{\varepsilon}$ are infinitesimal isotropic, i.e. $\lim_{\hat{\varepsilon} \to 0} \frac{\hat{\varepsilon}_{aug}}{\hat{\varepsilon}} = c$, where $c$ is a constant. Therefore, we can rewrite Eq. 4 as follows:

$$f(W(x + \hat{\delta})) = f(x) + c \cdot \hat{\varepsilon} + o(\hat{\varepsilon}). \tag{5}$$

Our goal is to prove that *distance (similarity) between AE and its neighbors can be significantly smaller (larger) than that of the clean sample in the space spanned by a SSL model*, which is equivalent to justify Eq. 6:

$$\|f(x_{adv}) - f(W(x_{adv}))\|_2^2 \geq \| \underbrace{f(x) - f(Wx)}_{\hat{\varepsilon}_{aug} = c \cdot \hat{\varepsilon}} \|_2^2. \tag{6}$$

Expending the left-hand item in Eq. 6, and defining $m = \nabla f(x)W\delta$, we can obtain the following.

$$\begin{aligned} \|f(x_{adv}) - f(Wx_{adv})\|_2^2 &= \|f(x + \delta) - f(W(x + \delta))\|_2^2 \\ &= \|f(x) + \nabla f(x)\delta - f(Wx) - \nabla f(Wx)W\delta\|_2^2 \\ &= \|\varepsilon - \nabla f(x)W\delta - o(\varepsilon)\|_2^2 = \|\varepsilon\|_2^2 + \|m\|_2^2 - 2|\langle \varepsilon, m \rangle| + o(\varepsilon) \end{aligned} \tag{7}$$

As mentioned in the prior literature (Mikołajczyk & Grochowski, 2018; Raff et al., 2019; Zeng et al., 2020), augmentations can effectively weaken adversarial perturbation $\delta$. Therefore, we assume that the influence caused by $W\delta$ is weaker than $\delta$ but stronger than the benign perturbation, $\hat{\delta}$. Formally, we have:

$$\| \underbrace{\nabla f(x)\delta}_{\varepsilon} \|_2 > \| \underbrace{\nabla f(x)W\delta}_{m} \|_2 > \| \underbrace{\nabla f(x)W\hat{\delta}}_{\hat{\varepsilon}_{aug} = c \cdot \hat{\varepsilon}} \|_2. \tag{8}$$

According to *Cauchy–Schwarz inequality* (Bhatia & Davis, 1995), we have the following chain of inequalities obtained by taking Eq. 8 into Eq. 7:

$$\|\varepsilon\|_2^2 + \|m\|_2^2 - 2|\langle \varepsilon, m \rangle| + o(\varepsilon) >$$
$$\|\varepsilon\|_2^2 + \|m\|_2^2 - 2\|\varepsilon\| \cdot \|m\| = (\|\varepsilon\|_2 - \|m\|_2)^2, \tag{9}$$

where $\|m\| \in (\|\hat{\varepsilon}_{aug}\|, \|\varepsilon\|)$ according to Eq. 8.

Finally, from Eq. 9 we observe that by applying proper data augmentation, the distance between AE and its neighbors in SSL's feature space $\|f(x_{adv}) - f(Wx_{adv})\|_2 = \|\|\varepsilon\|_2 - \|m\|_2\|_2$ can be significantly larger than that of clean samples $\|f(x) - f(Wx)\|_2 = o(\hat{\varepsilon})$. The enlarged distance is upper bounded by $\|\varepsilon\|_2/\|\hat{\varepsilon}_{aug}\|_2$ times that of the clean sample, which implies that the imperceptible perturbation $\delta$ in the image space can be significantly enlarged in SSL's feature space by referring to its neighbors. This exactly supports the design of BEYOND as described in Sec 2.1. In practice, we adopt various augmentations instead of a single type to generate multiple neighbors for AE detection, which reduces the randomness, resulting in more robust estimations.

## 3.2 ROBUSTNESS TO ADAPTIVE ATTACKS

**Adaptive Objective Loss Function.** Attackers can design adaptive attacks to try to bypass BE-YOND when the attacker knows all the parameters of the model and the detection strategy. For an SSL model with a feature extractor $f$, a projector $h$, and a classification head $g$, the classification branch can be formulated as $\mathbb{C} = f \circ g$ and the representation branch as $\mathbb{R} = f \circ h$. To attack effectively, the adversary must deceive the target model while guaranteeing the label consistency and representation similarity of the SSL model. Since BEYOND uses multiple augmentations, we estimate their impact on label consistency and representation similarity during the adaptive attack following Expectation over Transformation (EoT) (Athalye et al., 2018b) as:

$$Sim_l = \frac{1}{k}\sum_{i=1}^{k}\mathcal{L}\left(\mathbb{C}\left(W^i(x+\delta)\right), y_t\right), Sim_r = \frac{1}{k}\sum_{i=1}^{k}\mathcal{S}(\mathbb{R}(W^i(x+\delta)), \mathbb{R}(x+\delta)) \tag{10}$$

where $\mathcal{S}$ represents cosine similarity, $k$ represents the number of generated neighbors, and the linear augmentation function $W(x) = W(x, p)$; $p \sim P$ randomly samples $p$ from the parameter distribution $P$ to generate different neighbors. Note that we guarantee the generated neighbors are fixed each time by fixing the random seed. The adaptive adversaries perform attacks on the following objective function:

$$\min_{\delta} \mathcal{L}_C(x+\delta, y_t) + Sim_l - \alpha \cdot Sim_r, \tag{11}$$

where $\mathcal{L}_C$ indicates classifier's loss function, $y_t$ is the targeted class, and $\alpha$ refers to a hyperparameter[2], which is a trade-off parameter between label consistency and representation similarity. Experiments in the Appendix show that the adaptive attack is most effective when $\alpha = 1$.

**Conflicting Optimization Goals.** For an AE $x_{adv} = x + \delta$ and $y_{adv} = \mathbb{C}(x_{adv})$, the classification and representation outputs of its augmentation can be studied through their first-order Taylor expansion at $x$:

$$y_{aug} = \mathbb{C}(Wx_{adv}) = \mathbb{C}(Wx) + \nabla\mathbb{C}(Wx)W\delta$$
$$r_{aug} = \mathbb{R}(Wx_{adv}) = \mathbb{R}(Wx) + \nabla\mathbb{R}(Wx)W\delta \tag{12}$$

Since the SSL model is trained to generate close representations between a sample and its augmentation ($\mathbb{C}(Wx) \approx \mathbb{C}(x), \mathbb{R}(Wx) \approx \mathbb{R}(x)$), the differences of label and representation between the original sample and its augmentation are denoted as:

$$y_{aug} - y \approx \nabla\mathbb{C}(x)W\delta, r_{aug} - r \approx \nabla\mathbb{R}(x)W\delta \tag{13}$$

Therefore, to ensure the label consistency of AE, i.e., $y_{aug} = y_t \neq y$, the optimization goal of the adaptive attack is making $\delta$ larger within the perturbation budget:

$$\delta = \max_{\|\delta\| \leq \epsilon} (\nabla\mathbb{C}(x)W\delta) \tag{14}$$

---

[2]Note that we employ cosine metric that is negatively correlated with the similarity, so that the $Sim_r$ item is preceded by a minus sign.

| AUC (%) | Unseen: Attacks used in training are preclude from tests. | | | | Seen: Attacks used in training are included in tests. | | | | |
|---|---|---|---|---|---|---|---|---|---|
| | FGSM | PGD | AutoAttack | Square | FGSM | PGD | CW | AutoAttack | Square |
| DkNN | $61.55_{\pm 0.023}$ | $51.22_{\pm 0.026}$ | $52.12_{\pm 0.023}$ | $59.46_{\pm 0.022}$ | $61.55_{\pm 0.023}$ | $51.22_{\pm 0.026}$ | $61.52_{\pm 0.028}$ | $52.12_{\pm 0.023}$ | $59.46_{\pm 0.022}$ |
| kNN | $61.83_{\pm 0.018}$ | $54.52_{\pm 0.022}$ | $52.67_{\pm 0.022}$ | $73.39_{\pm 0.02}$ | $61.83_{\pm 0.018}$ | $54.52_{\pm 0.022}$ | $62.23_{\pm 0.019}$ | $52.67_{\pm 0.022}$ | $73.39_{\pm 0.02}$ |
| LID | $71.08_{\pm 0.024}$ | $61.33_{\pm 0.025}$ | $55.56_{\pm 0.021}$ | $66.18_{\pm 0.025}$ | $73.61_{\pm 0.02}$ | $67.98_{\pm 0.02}$ | $55.68_{\pm 0.021}$ | $56.33_{\pm 0.024}$ | $85.94_{\pm 0.018}$ |
| Hu | $84.51_{\pm 0.025}$ | $58.59_{\pm 0.028}$ | $53.55_{\pm 0.029}$ | $95.82_{\pm 0.02}$ | $84.51_{\pm 0.025}$ | $58.59_{\pm 0.028}$ | $91.02_{\pm 0.022}$ | $53.55_{\pm 0.029}$ | $95.82_{\pm 0.02}$ |
| Mao | $95.33_{\pm 0.012}$ | $82.61_{\pm 0.016}$ | $81.95_{\pm 0.02}$ | $85.76_{\pm 0.019}$ | $95.33_{\pm 0.012}$ | $82.61_{\pm 0.016}$ | $83.10_{\pm 0.018}$ | $81.95_{\pm 0.02}$ | $85.76_{\pm 0.019}$ |
| LNG | $98.51$ | $63.14$ | $58.47$ | $94.71$ | $99.88$ | $91.39$ | $89.74$ | $84.03$ | $98.82$ |
| BEYOND | $98.89_{\pm 0.013}$ | $99.28_{\pm 0.02}$ | $99.16_{\pm 0.021}$ | $99.27_{\pm 0.016}$ | $98.89_{\pm 0.013}$ | $99.28_{\pm 0.02}$ | $99.20_{\pm 0.008}$ | $99.16_{\pm 0.021}$ | $99.27_{\pm 0.016}$ |

Table 1: The AUC of Different Adversarial Detection Approaches on CIFAR-10. The results are the mean and standard deviation of 5 runs. LNG is not open-sourced and the data comes from its report. To align with baselines, classifier: ResNet110, FGSM: $\epsilon = 0.05$, PGD: $\epsilon = 0.02$. Note that **BEYOND needs no AE for training**, leading to the same value on both *seen* and *unseen* settings. The **bolded** values are the best performance, and the *underlined italicized* values are the second-best performance, the same below.

Conversely, the optimization goal of representation similarity ($r_{aug} = r$) is making $\delta$ smaller:

$$\delta = min(\nabla \mathbb{R}(x) W \delta) \tag{15}$$

The conflicting optimization goals cause the gradients generated by attack label consistency and representation similarity to cancel each other out, which is the reason why BEYOND is robust to adaptive attacks. Fig. 11 visualizes the gradient cancellation of the two optimization objectives.

Moreover, the above analysis demonstrates that small perturbations do not guarantee label consistency for AEs, while large perturbations impair representation similarity, which is consistent with the empirical results in Sec 4.4.

## 4 EVALUATION

### 4.1 EXPERIMENTAL SETTING

**Limited knowledge attack & Perfect knowledge attack.** Following (Apruzzese et al., 2023), in the limited knowledge attack setting, the adversary has complete knowledge of the classifier, while the detection strategy is confidential. Whereas in an adaptive attack (perfect knowledge) setting, the adversary is aware of the detection strategy.

**Datasets & Target models.** We conduct experiments on three commonly adopted datasets including CIFAR-10 (Krizhevsky et al., 2009), CIFAR-100, and IMAGENET (Krizhevsky et al., 2012) The details of the target models (classifiers), and the employed SSL models together with their original classification accuracy on clean samples are summarized in Tab. 6 [3].

**Augmentations.** The types of augmentation used by BEYOND to generate neighbors are consistent with Sim-Siam, including horizontal flipping, cropping, color jitter, and greyscale. However, BEYOND fixes the random seed to prevent benefiting from randomization. We generate 50 neighbors for each sample, and the ablation study on the number of neighbors is further discussed in Sec. 4.4.

**Attacks.** Evaluations of limited knowledge attacks are conducted on FGSM, PGD, C&W, and AutoAttack. AutoAttack includes APGD, APGD-T, FAB-T, and Square (Andriushchenko et al., 2020), where APGD-T and FAB-T are targeted attacks and Square is a black-box attack. As for adaptive attacks, we employed the most adopted EoT and Orthogonal-PGD, which is a recent adaptive attack designed for AE detectors.

**Metrics.** Following previous work (Abusnaina et al., 2021), we employ ROC curve & AUC and Robust Accuracy (RA) as evaluation metrics.

- **ROC curve & AUC:** Receiver Operating Characteristic (ROC) curves describe the impact of various thresholds on detection performance, and the Area Under the Curve (AUC) is an overall indicator of the ROC curve.

- **Robust Accuracy (RA)**: We employ RA as an evaluation metric, which can reflect the overall system performance against adaptive attacks by considering both the classifier and the detector.

---

[3]The pre-trained SSL models for CIFAR-10 and CIFAR-100 are from Solo-learn (da Costa et al., 2022), and for IMAGENET are from SimSiam (Chen & He, 2021).

| AUC (%) | Unseen | | Seen | | |
|---|---|---|---|---|---|
| | **FGSM** | **PGD** | **FGSM** | **PGD** | **CW** |
| DkNN | $89.16_{\pm0.038}$ | $78.00_{\pm0.041}$ | $89.16_{\pm0.038}$ | $78.00_{\pm0.041}$ | $68.91_{\pm0.044}$ |
| kNN | $51.63_{\pm0.04}$ | $51.14_{\pm0.039}$ | $51.63_{\pm0.04}$ | $51.14_{\pm0.039}$ | $50.73_{\pm0.04}$ |
| LID | $90.32_{\pm0.046}$ | $52.56_{\pm0.038}$ | $\underline{99.24}_{\pm0.043}$ | $\underline{98.09}_{\pm0.042}$ | $58.83_{\pm0.041}$ |
| Hu | $72.56_{\pm0.037}$ | $86.00_{\pm0.042}$ | $72.56_{\pm0.037}$ | $86.00_{\pm0.042}$ | $80.79_{\pm0.044}$ |
| LNG | $\underline{96.85}$ | $\underline{89.61}$ | **99.53** | **98.42** | $\underline{86.05}$ |
| BEYOND | $\mathbf{97.59}_{\pm0.04}$ | $\mathbf{96.26}_{\pm0.045}$ | $97.59_{\pm0.04}$ | $96.26_{\pm0.045}$ | $\mathbf{95.46}_{\pm0.047}$ |

Table 2: The AUC of Different Adversarial Detection Approaches on IMAGENET. To align with baselines, classifier: DenseNet121, FGSM: $\epsilon = 0.05$, PGD: $\epsilon = 0.02$. Due to memory and resource constraints, baseline methods are not evaluated against AutoAttack on IMAGENET.

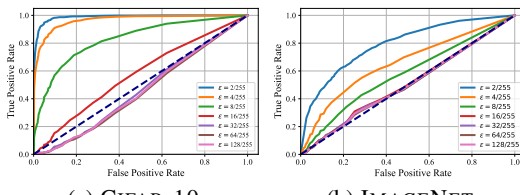

| Model | RA | | Acc. on clean samples | |
|---|---|---|---|---|
| | ATC | ATC+BEYOND | ATC | ATC+BEYOND |
| (Rebuffi et al., 2021) | 66.20% | **84.40%** | 92.23% | **92.83%** |
| (Gowal et al., 2021) | 64.10% | **81.50%** | 88.74% | **90.81%** |
| (Gowal et al., 2020) | 64.70% | **83.80%** | 91.10% | **91.79%** |
| (Rebuffi et al., 2021) | 62.20% | **81.30%** | 88.50% | **90.51%** |

(a) CIFAR-10  (b) IMAGENET

Figure 3: ATC+BEYOND against AutoAttack on CIFAR-10.

Figure 2: ROC Curve of BEYOND against adaptive attacks with different perturbation budgets.

**Baselines.** We choose five detection-based defense methods as baselines: kNN (Dubey et al., 2019), DkNN (Papernot & McDaniel, 2018), LID (Ma et al., 2018), (Hu et al., 2019) and LNG, which also consider the relationship between the input and its neighbors to some extent. (Mao et al., 2021) trains self-supervised branches to purify the adversarial examples, which is one of the best adaptive robust methods available.

## 4.2 DEFENDING LIMITED KNOWLEDGE ATTACKS

We compare the AUC of BEYOND with DkNN, kNN, LID, Hu, Mao, and LNG on CIFAR-10 and IMAGENET. Since LID and LNG rely on reference AEs, we report detection performance on both seen and unseen attacks. In the seen attack setting, LID and LNG are trained with all types of attacks, while using only the C&W attack in the unseen attack setting. Note that the detection performance for seen and unseen attacks is consistent for detection methods without AEs training.

Tab. 1 shows that BEYOND consistently outperforms SOTA AE detectors on CIFAR-10, and the performance advantage is particularly significant when detecting unseen attacks. This is because BEYOND uses the augmentations of the input as its neighbors without relying on prior adversarial knowledge. And according to the conclusion in Sec 3, adversarial perturbations impair label consistency and representation similarity, which enables BEYOND to distinguish AEs from benign ones with high accuracy.

Tab. 2 compares the AUC scores of BEYOND with SOTA AE detectors on IMAGENET. Experimental results show that BEYOND outperforms the SOTA AE detectors in detecting unseen attacks. For seen attacks, because BEYOND does not rely on prior adversarial knowledge, the detection performance against FGSM and PGD is slightly lower than that of LNG. While for stronger attacks, i.e, C&W, BEYOND outperforms baselines by a significant margin. For more information about BEYOND's detection performance (TPR@FPR) on CIFAR-10, CIFAR-100 and IMAGENET, please refer to the Appendix.

**Improved Robustness with ATC.** As a plug-and-play approach, BEYOND integrates well with existing Adversarial Trained Classifier (ATC)[4]. Tab. 3 shows the accuracy on clean samples and RA against AutoAttack of ATC combined with BEYOND on CIFAR-10. As can be seen the addition of BEYOND increases the robustness of ATC by a significant margin on both clean samples and AEs. Note that the Acc in Table 3 is defined in Appendix.

---

[4]All ATCs are sourced from RobustBench (Croce et al., 2020).

| Classifier | Method | RA |
|---|---|---|
| Standard | Mao | 18.97% |
| | BEYOND | 19.45% |
| ATC | Mao | 75.09% |
| | BEYOND | 93.20% |

Table 3: Comparison of robust accuracy against adaptive attacks on CIFAR-10.

| Defense | $L_\infty$=0.01 | | $L_\infty$=8/255 | |
|---|---|---|---|---|
| | RA@FPR5% | RA@FPR50% | RA@FPR5% | RA@FPR50% |
| **BEYOND** | *88.38%* | *98.81%* | *13.80%* | *48.20%* |
| **BEYOND +ATC** | **96.30%** | **99.30%** | **94.50%** | **97.80%** |
| Trapdoor (Shan et al., 2020) | 0.00% | 7.00% | 0.00% | 8.00% |
| DLA (Sperl et al., 2020) | 62.60% | 83.70% | 0.00% | 28.20% |
| SID (Tian et al., 2021) | 6.90% | 23.40% | 0.00% | 1.60% |
| SPAM (Liu et al., 2019) | 1.20% | 46.00% | 0.00% | 38.00% |

Table 4: Robust Accuracy under Orthogonal-PGD Attack.

## 4.3 DEFENDING ADAPTIVE ATTACKS

**ROC Curve of Perturbation Budgets.** Fig. 2 summarizes the ROC curve varying with different perturbation budgets on CIFAR-10 and IMAGENET. Our analysis regarding Fig. 2 is as follows: 1) BEYOND can be bypassed when perturbations are large enough, due to large perturbations circumventing the transformation. This proves that BEYOND is not gradient masking (Athalye et al., 2018a) and our adaptive attack design is effective. However, large perturbations are easier to perceive. 2) When the perturbation is small, the detection performance of BEYOND for adaptive attacks still maintains a high level, because small perturbations cannot guarantee both label consistency and representation similarity (as shown in Fig. 6 (a)). The above empirical conclusions are consistent with the analysis in Sec 3.2.

**Performance against Orthogonal-PGD Adaptive Attacks.** Orthogonal-Projected Gradient Descent (Orthogonal-PGD) is a recently proposed AE detection benchmark for adaptive attacks. Orthogonal-PGD has two attack strategies: orthogonal strategy and selection strategy. Tab. 4 shows BEYOND outperforms the four baselines by a considerable margin in orthogonal strategy, especially under small perturbations. For the worst case, BEYOND can still keep 13.8% ($L_\infty = 8/255$). Furthermore, incorporating ATC can significantly improve the detection performance of BEYOND against large perturbation to 94.5%. See the Appendix for more selection strategy results. In addition, the coupling of the classifier and defense model in Mao's method is not consistent with the Orthogonal-PGD setting. We compare the robust accuracy of BEYOND and Mao for general adaptive attacks in Tab. 3, which shows that BEYOND outperforms Mao et al. against adaptive attacks with both standard classifier and ATC.

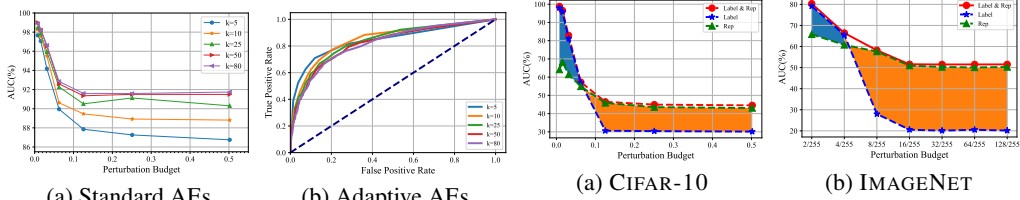

| (a) Standard AEs | (b) Adaptive AEs | (a) CIFAR-10 | (b) IMAGENET |

Figure 4: Ablation Study of the Number of Neighbors.

Figure 5: Ablation studies of representation similarity & label consistency against adaptive attacks.

## 4.4 ABLATION STUDY

**The Number of Neighbors K.** To study the impact of the number of neighbors against standard and adaptive attacks, we conduct BEYOND with $k = 5, 10, 25, 50, 80$. Fig. 4 (a) shows the effect of the number of neighbors on the detection performance against PGD with different perturbation budgets. It can be observed that the detection performance with a large number of neighbors is better, but the performance is not significantly improved when the number of neighbors exceeds 50. As for adaptive attacks, Fig. 4 (b) shows the performance of the adaptive attacks generated for different $k$ ($\epsilon = 8/255$). Contrary to intuition, adaptive attacks perform slightly worse when $k$ is small. This is because only four linear transformations (horizontal flip, crop, color jitter, and grayscale) are deployed in BEYOND, and different numbers of neighbors just used different transformation parameters. When $k$ is small, the difference between neighbors is great, and the optimization of adaptive attack is difficult (multi-task learning increases model robustness (Mao et al., 2020)); when $k$ is large, there may be similar neighbors that provide rich information for adaptive attacks for each transform, which is shown in the Appendix.

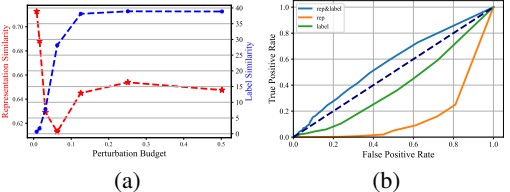

| | Model | FLOPs(G) | Params(M) | Time(s) | Overall |
|---|---|---|---|---|---|
| **ATC** | (Rebuffi et al., 2021) | 38.8 | 254.44 | 1.21 | 11945 |
| | (Gowal et al., 2021) | 38.8 | 254.44 | 1.21 | 11945 |
| | (Gowal et al., 2020) | 38.8 | 254.44 | *1.21* | 11945 |
| | (Rebuffi et al., 2021) | 60.57 | 396.23 | 1.24 | 29760 |
| **Det.** | Mao | 5.25 | 38.12 | 38.46 | 7697 |
| | LNG | **0.286** | **8.33** | 9.22 | *20.521* |
| | BEYOND | *0.715* | *20.62* | **1.12** | **16.512** |

Figure 6: Trade-off between Label Consistency and Representation Similarity.

Table 5: Comparison of Implementation Costs.

**Contribution of Representation Similarity & Label Consistency against Adaptive Attacks.** The analysis in Sec. 3.2 shows label consistency is more beneficial for detecting small perturbations, while representational similarity is favourable for large perturbations, which is consistent with results in Fig. 5. When the perturbation is small, the detection performance based on label consistency (blue line) is better than representation similarity (green line). As perturbation increases, representation similarity is difficult to maintain, leading to higher performance of representation similarity-based detectors. In summary, the label consistency and representation similarity have different sensitivities to perturbation, so their cooperation has the optimal performance (red line).

**Representation Similarity & Label Consistency.** The previous analysis and empirical results have proved that there is a trade-off between label consistency and representation similarity. Fig. 6 (a) shows the variation of label consistency and representation similarity with perturbation size on CIFAR-100. We can observe that label consistency and representation similarity respond differently to the perturbation size, small perturbations are beneficial for representation similarity, and large perturbations favor label consistency, which matches the conclusion in Sec. 3.2. Furthermore, both objectives can be optimized simultaneously when the perturbation is large enough, which is why the adaptive attack in Fig. 2 can completely break BEYOND when the perturbation budget is larger than $16/255$. Fig. 6 (b) shows that when there is only one detection strategy, either label consistency and representation similarity, the adaptive attack can break through the defense. However, when attacking both strategies, the attack performance decreases. Hence, the robustness of BEYOND to adaptive attacks comes from the conflicts arising from optimizing these two strategies. See Appendix for more visualization results of optimization conflicts.

### 4.5 IMPLEMENTATION COSTS

BEYOND uses an additional SSL model for AE detection, which inevitably increases the computational and storage cost. And the inference time (speed) is also considered in practice. Tab. 5 presents the comparison for SOTA adversarial training defense and AE detection method, i.e. LNG. The detection models have a smaller model structure than those of ATCs, which can be reflected by the *Params* and *FLOPs* (Xie et al., 2020) being much lower than those of ATC. For BEYOND, the projection head is a three-layer FC, leading to higher parameters and *FLOPs* than LNG. However, BEYOND only compares the relationship between neighbors without calculating the distance with the reference set, resulting in a faster inference speed than that of LNG. The method of Mao et al. requires iteration, making its inference time unaffordable (Croce et al., 2022). We show the FLOPs $\times$ Params $\times$ Time as the *Overall* metric in Tab. 5's last column for overall comparison. If cost is a real concern in some scenarios, we can further reduce the cost with some strategy, e.g., reducing the neighbor number, without compromising performance significantly, as shown in Fig. 4 (a).

### 5 CONCLUSION

In this paper, we take the first step to detect AEs by identifying abnormal relations between AEs and their neighbors without prior knowledge of AEs. Samples that have low label consistency and representation similarity with their neighbors are detected as AE. Experiments with limited and perfect knowledge attacks show that BEYOND outperforms the SOTA AE detectors in both detection ability and efficiency. Moreover, as a plug-and-play model, BEYOND can be well integrated with ATC to further improve robustness.

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

## A    APPENDIX

## A    DATASETS & MODELS

We conduct experiments on three commonly adopted datasets including CIFAR-10 Krizhevsky et al. (2009), CIFAR-100, and a more IMAGENET Krizhevsky et al. (2012). The details of the target models (classifiers), and the employed SSL models together with their original classification accuracy on clean samples are summarized in Tab. 6 [5].

| Dataset | Classifier SSL | Acc. on clean samples↑ | |
|---|---|---|---|
| | | Classifier | SSL |
| CIFAR-10 | ResNet18 | 91.53% | 90.74% |
| CIFAR-100 | ResNet18 | 75.34% | 66.04% |
| IMAGENET | ResNet50 | 80.86% | 68.30% |

Table 6: Information of datasets and models.

## B    DETECTION PERFORMANCE

### B.1    TPR@FPR AGAINST LIMITED KNOWLEDGE ATTACKS.

Tab. 7 reports TPR@FPR5% to show the AE detection performance of BEYOND. It can be observed that BEYOND maintains a high detection performance on various attacks and datasets, which is attributed to our detection mechanism. Combining label consistency and representation similarity, BEYOND identifies AEs without reference AE set.

---

[5]The pre-trained SSL models for CIFAR-10 and CIFAR-100 are from Solo-learn da Costa et al. (2022), and for IMAGENET are from SimSiam Chen & He (2021).

| Dataset | CIFAR-10 | CIFAR-100 | IMAGENET |
|---|---|---|---|
| Attack | TPR@FPR5% ↑ | | |
| FGSM | 86.16% | 89.80% | 61.05% |
| PGD | 82.80% | 85.90% | 89.80% |
| C&W | 91.48% | 91.96% | 76.69% |
| AutoAttack | 93.42% | 90.90% | 84.25% |

Table 7: TPR@FPR 5% of BEYOND against Limited Knowledge Attacks. All attacks are performed under $L_\infty = 8/255$.

Tab. 8 reports TPR@FPR 3% to further demonstrate the AE detection capability of BEYOND. Because the detection mechanism does not rely on additional prior knowledge of AE or model retraining, it has been confirmed that BEYOND can generalize well to defend various attacks. Furthermore, on the complex dataset, i.e., IMAGENET, BEYOND still maintains a high detection performance.

| Dataset | CIFAR-10 | CIFAR-100 | IMAGENET |
|---|---|---|---|
| Attack | TPR@FPR3% ↑ | | |
| FGSM | 76.37% | 81.93% | 51.74% |
| PGD | 69.50% | 76.00% | 82.20% |
| C&W | 85.29% | 84.32% | 68.50% |
| AutoAttack | 88.33% | 83.91% | 72.06% |

Table 8: TPR@FPR 3% of BEYOND against Limited Knowledge Attacks. All attacks are performed under $L_\infty = 8/255$.

## B.2 ACCURACY WITH ATC

Following (Yang et al., 2022), Accuracy in Table 3 indicates the detector's accuracy on clean samples by combining the detector with the classifier, and calculated as follows:

$$Acc = \frac{\#\text{Classifier correct\&Detector pass}}{\#\text{All clean samples}} + \frac{\#\text{Classifier wrong\&Detector reject}}{\#\text{All clean samples}}$$

## B.3 PERFORMANCE AGAINST ORTHOGONAL-PGD SELECTION STRATEGY ADAPTIVE ATTACKS

Orthogonal-Projected Gradient Descent (Orthogonal-PGD) is a recently proposed AE detection benchmark. In the selection strategy, Orthogonal-PGD updates the input by selectively exploiting perturbations produced by either the classifier or the detector to avoid perturbation waste. Tab. 9 shows BEYOND outperforms the four baselines by a considerable margin in selection strategy, especially under small perturbations.

For the worst case, BEYOND can still maintain 8.04% ($L_\infty = 8/255$), while the baselines are only 0.4%. Furthermore, incorporating ATC can significantly improve the detection performance of BEYOND against large perturbation to 91.5%.

## B.4 DETECTION PERFORMANCE ON CIFAR-100

Fig. 7 shows the detection performance of BEYOND against adaptive attacks on CIFAR-100 and the contribution of label consistency and representation similarity. It can be seen BEYOND is effective for detecting adaptive attacks on CIFAR-10. Meanwhile, label consistency is more suitable for detecting small perturbations, while representation similarity is favourable for large perturbations, which is consistent with the conclusion on CIFAR-10 and IMAGENET.

| Defense | $L_\infty$=0.01 | | $L_\infty$=8/255 | |
|---|---|---|---|---|
| | RA @FPR5% | RA @FPR50% | RA @FPR5% | RA @FPR50% |
| **BEYOND** | *79.63%* | *97.47%* | *8.04%* | *40.42%* |
| **BEYOND +ATC** | **95.80%** | **99.40%** | **91.50%** | **95.90%** |
| Trapdoor | 0.20% | 49.50% | 0.40% | 37.20% |
| DLA'20 | 17.00% | 55.90% | 0.00% | 13.50% |
| SID'21 | 8.90% | 50.90% | 0.00% | 11.40% |

Table 9: Robust Accuracy under Orthogonal-PGD selection strategy on CIFAR-10. The **bolded** values are the best performance and the *underlined italicized* values are the second-best performance.

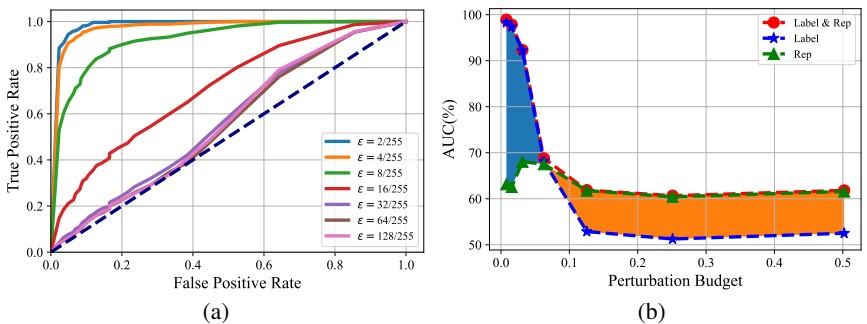

(a)  (b)

Figure 7: (a) Detection performance against adaptive attacks on CIFAR-100. (b) Contribution of label consistency and representation similarity on CIFAR-100

## B.5   DETECTION PERFORMANCE FOR VARIOUS TYPES OF ATTACKS

To evaluate the detection performance of BEYOND for different types of attacks, we test the most representative method that supports multiple norm attacks, AutoAttack. AutoAttack supports $L_\infty$, $L_2$ and $L_1$ norm attacks. In the main paper, we only report the detection performance of BEYOND against AutoAttack $L_\infty$. Table 11 shows the performance of BEYOND against AutoAttack with different norms. Where the perturbation budgets ($\epsilon$) on CIFAR-10 are $8/255$ ($L_\infty$), $0.5$ ($L_2$), and $8$ ($L_1$); and on IMAGENET are $8/255$ ($L_\infty$), $3$ ($L2$), and $64$ ($L_1$). The results show BEYOND is still effective against attacks based on different norms.

## B.6   HYPERPARAMETER ALPHA IN ADAPTIVE ATTACKS

The design of the adaptive attack in Eq. 11 includes a hyperparameter $\alpha$, which is a trade-off parameter between label consistency and representation similarity. Tab. 10 shows the AUC of BEYOND under different $\alpha$. As shown, when $\alpha = 0$, i.e. the attacker only attacks the label consistency detection mechanism, the AUC score is still high, which proves that our approach is not based on the weak transferability of AEs. Moreover, adaptive attacks are strongest when $\alpha = 1$, which is used for all tests.

## C   ABLATION STUDY ON SSL MODEL

BEYOND can flexibly cooperate with various SSL models without compromising AE detection performance, as long as the SSL model is trained to generate similar representations for the input and its augmentations. Without loss of generality, we let BEYOND encapsulate five different SSL models, including: SimSiam (Chen & He, 2021) (employed in main paper), BYOL (Grill et al., 2020), MoCov3 (Chen et al., 2021), SwAV (Caron et al., 2020b), DeepClusterv2 (Caron et al., 2020a), and report the BEYOND's AE detection performance in Tab. 12. As can be seen, BEYOND generalizes well with all five SSL models under various experimental settings.

| $\alpha$ | 0 | 1 | 10 | 20 | 50 | 100 |
|---|---|---|---|---|---|---|
| CIFAR-10 | 82.03% | 63.91% | 64.57% | 76.15% | 88.56% | 92.53% |
| CIFAR-100 | 90.58% | 88.49% | 91.61% | 93.10% | 94.05% | 94.37% |

Table 10: AUC for Adaptive Attack under different $\alpha$.

| AUC(%) | $L_\infty$ | $L_2$ | $L_1$ |
|---|---|---|---|
| CIFAR-10 | 99.18 | 99.13 | 99.07 |
| IMAGENET | 97.14 | 97.26 | 97.18 |

Table 11: Detection performance of BEYOND against AutoAttack with different norms.

## D   DISPLAY OF GENERATED NEIGHBORS

Fig. 8 shows the 50 neighbors augmented by the original image. Augmentations are made up of four linear variations including color jitter, crop, horizontal flip and greyscale. Neighbors are generated by random combinations of transformation parameters, whose consistency is ensured by fixing random seeds. It can be noticed that when the number of generated neighbors is small, there is a large difference between neighbors, while when the number of generated neighbors is large, there are similar neighbors. This may be the reason why the adaptive attack is a little more difficult to break BEYOND when $k$ is small in Fig. 4.

## E   SELECT EFFECTIVE AUGMENTATIONS

To better improve the effectiveness of BEYOND, we analyze the conditions under which the augmentation can effectively weaken adversarial perturbation. Effective data augmentation makes the augmented label $y_{aug}$ tend to the ground-truth label $y_{true}$ and away from the adversarial label $y_{adv}$:

$$||y_{aug} - y_{true}||_2 \le ||y_{aug} - y_{adv}||_2 \le ||y_{adv} - y_{true}||_2. \tag{16}$$

Since $y_{true}$ is the one-hot encoding, the range of $||y_{adv} - y_{true}||_2$ is $(\sqrt{2}/2, \sqrt{2})$. The distance is $\sqrt{2}$ when the item corresponding to $y_{adv}$ is 1 in the logits of $y_{adv}$, and $\sqrt{2}/2$ when the item corresponding to $y_{adv}$ and $y_{true}$ both occupy 1/2. Given a SSL-based classifier, $C$, we have:

$$C(W(x + \delta)) = C(Wx) + \nabla C(Wx)W\delta$$
$$= y_{true} + \nabla C(Wx)W\delta = y_{aug}. \tag{17}$$

Therefore, the distance between $y_{aug}$ and $y_{true}$ is:

$$||y_{aug} - y_{true}||_2 = ||\nabla C(Wx)W\delta||_2$$
$$\le ||\nabla C(Wx)W||_2||\delta||_2 \le ||\nabla C(Wx)W||_2\epsilon \tag{18}$$

where $||\delta||_2$ is bounded by $\epsilon$. Eq. 16 always holds, then:

$$||\nabla C(Wx)W||_2\epsilon \le \frac{\sqrt{2}}{2} \Rightarrow ||\nabla C(Wx)W||_2 \le \frac{\sqrt{2}}{2\epsilon}. \tag{19}$$

In summary, augmentation can mitigate adversarial perturbation when it satisfies Eq. 19.

To further validate our analysis, we generate 1000 adversarial examples by PGD with $\epsilon = 8/255$ on CIFAR-10. Table 13 shows the ratios for different data augmentations meeting the threshold $\frac{\sqrt{2}}{2\epsilon}$. A higher ratio means the augmentation is more effective. It can be observed that Rotation, Color Jitter and Compose are the three most effective augmentations according to our analysis. To further validate our analysis, we perform t-sne (Van der Maaten & Hinton, 2008) visualizations of the SSL representations of clean and AEs processed by different augmentation methods. We utilize a self-supervised feature extractor and projection head to obtain SSL representations and use augmentation methods to generate 20 neighbors for both clean samples and AEs. As seen in Fig. 10, the effective augmentation methods with the high ratio in Table 13 can effectively increase the distance between AEs and their neighbors. For example, Rotation has the highest ratio in Table 13, and the distance between AE and its neighbors in Fig. 10 is larger than that of clean samples. While Horizontal and Vertical have the lowest ratio, and the distance between AE and its neighbors is still close in Fig. 10

| Dataset | Model | FGSM | PGD | C&W | APGD-CE | APGD-T | FAB-T | Square |
|---|---|---|---|---|---|---|---|---|
| **CIFAR-10** | SimSiam | 97.17% | 96.48% | 98.22% | 96.60% | 99.45% | 99.14% | 98.60% |
| | BYOL | 97.22% | 94.60% | 98.38% | 94.97% | 99.54% | 99.61% | 99.02% |
| | MoCo v3 | 98.54% | 98.26% | 99.25% | 98.38% | 99.82% | 99.69% | 99.31% |
| | SwAV | 96.29% | 94.81% | 97.62% | 95.40% | 99.14% | 98.73% | 98.16% |
| | DeepCluster v2 | 92.68% | 89.28% | 95.32% | 90.72% | 98.04% | 97.56% | 96.55% |
| **CIFAR-100** | SimSiam | 97.82% | 97.29% | 97.93% | 97.40% | 98.33% | 97.99% | 97.80% |
| | BYOL | 98.04% | 97.00% | 98.01% | 96.75% | 98.45% | 98.33% | 98.13% |
| | MoCo v3 | 98.34% | 98.10% | 98.50% | 98.14% | 98.81% | 98.58% | 98.44% |
| | SwAV | 97.58% | 96.91% | 97.85% | 97.01% | 98.44% | 97.94% | 97.70% |
| **IMAGENET** | SimSiam | 92.01% | 96.88% | 94.56% | 97.15% | 97.45% | 95.47% | 94.58% |
| | BYOL | 92.01% | 96.57% | 94.58% | 96.67% | 97.00% | 95.65% | 94.25% |

Table 12: AUC scores for BEYOND with various SSL models against 7 adversarial attacks. SSL models trained on CIFAR-10 and CIFAR-100 are implemented with ResNet18, trained on IMA-GENET are implemented with ResNet50.

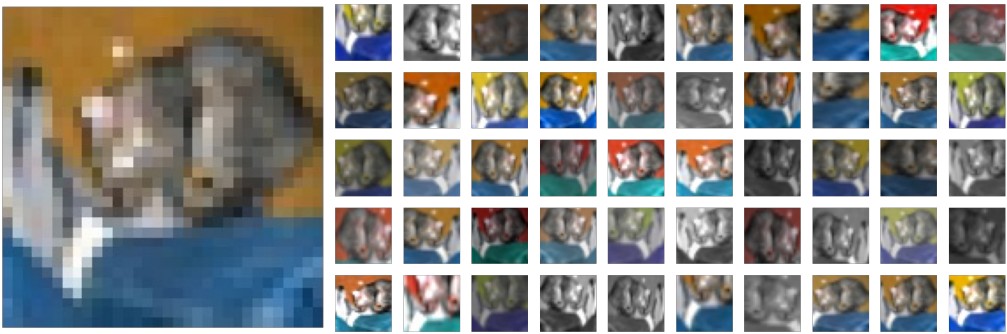

Figure 8: Display of generated neighbors. The original image is on the left and the generated 50 neighbors are on the right.

Moreover, we test the detection performance of high ratio augmentations and low ratio augmentations in Table 14. It can be seen that the average detection performance of the effective augmentations obtained by our analysis is 5% higher than that of the other augmentations.

## F  CONFLICT RATE OF LABEL CONSISTENCY AND REPRESENTATION SIMILARITY

The conflict between label consistency and representation similarity stems from their different optimization goal. Fig. 9 shows the gradient conflict rate for adaptive attacks with different step sizes on different perturbation budgets. We can find that the gradient conflict rate decreases for large perturbations and converges as the perturbation further increases, with the convergence point being consistent with the turning point in Fig. 6 of the main paper.

Sec. 3.2 demonstrates the conflict between label consistency and representation similarity stems from their different optimization goals. Fig. 11 visualizes the gradients produced by optimizing label consistency and representation similarity on the input. It's shown that attacks on label consistency or representation similarity produce gradients that modify the input in a certain direction, but optimizing for both leads to conflicting gradients. The experiments in the Appendix show that the gradient conflict rate decreases when the perturbation becomes larger, which is consistent with the results in Fig. 6 (a).

## G  RELATED WORKS

The authors in (Szegedy et al., 2013) first discovered that an adversary could maximize the prediction error of the network by adding some imperceptible perturbation, $\delta$, which is typically bounded by a perturbation budget, $\epsilon$, under an $L_p$-norm, e.g., $L_\infty$ and $L_2$. Project Gradient Descent (PGD) pro-

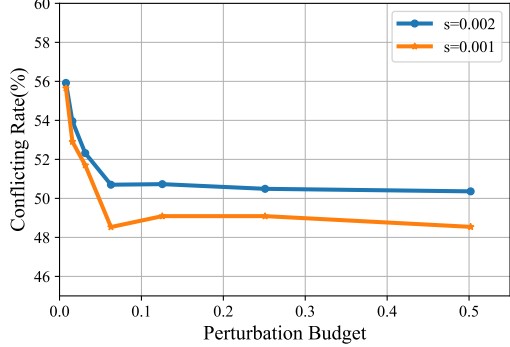

Figure 9: Conflicting rate for optimizing label consistency and representation similarity with different attack step size.

| Aug | Ratio | Aug | Ratio |
|---|---|---|---|
| Rotation | 99.9% | Vertical | 25.9% |
| Crop | 40.7% | Color Jitter | 99.0% |
| Resize | 74.0% | Gray | 40.6% |
| Horizontal | 25.9% | Compose | 99.7% |

Table 13: The ratio of different data augmentations meeting the threshold. Compose is a combination of augmentations used to train SSL models, including crop, resize, horizontal flip, and color jitter.

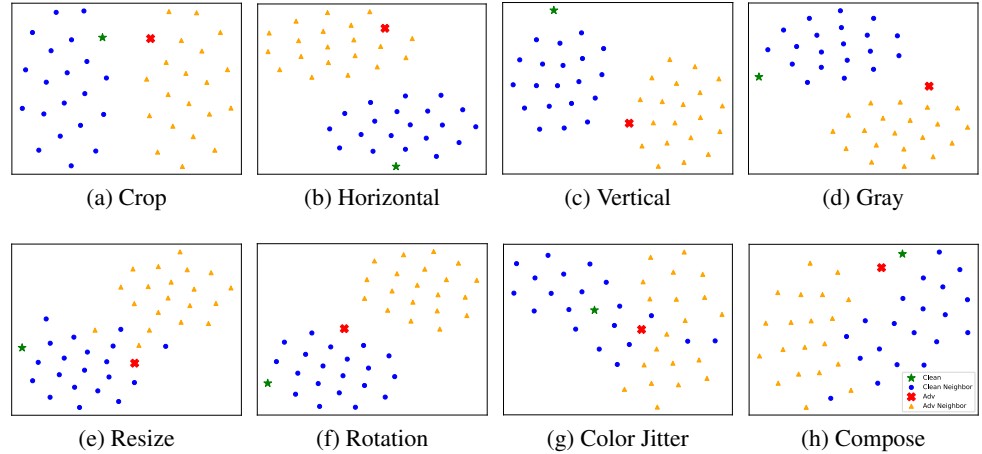

(a) Crop  (b) Horizontal  (c) Vertical  (d) Gray

(e) Resize  (f) Rotation  (g) Color Jitter  (h) Compose

Figure 10: Visualization of clean sample and AE with different augmented neighborhoods.

posed by (Madry et al., 2017) is one of the most powerful iterative attacks. PGD motivates various gradient-based attacks such as AutoAttack (Croce & Hein, 2020) and Orthogonal-PGD (Bryniarski et al., 2021), which can break many SOTA AE defenses (Croce et al., 2022). Another widely adopted adversarial attack is C&W (Carlini & Wagner, 2017). Compared to the norm-bounded PGD attack, C&W conducts AEs with a high attack success rate by formulating the adversarial attack problem as an optimization problem.

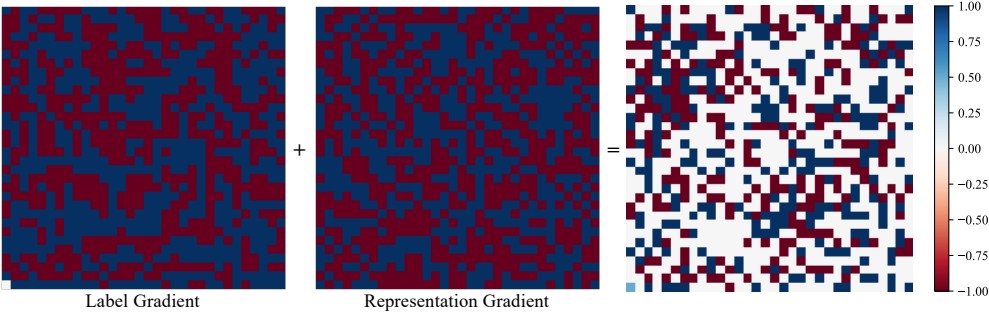

Figure 11: Gradient conflict between label consistency and representation similarity. The colored pixels represent the gradient direction, while the blank means gradient conflict.

| Augmentation | FGSM | PGD | CW | AutoAttack | Average |
|---|---|---|---|---|---|
| ColorJitter&Resize&Rotation | 97.11% | 96.55% | 98.15% | 96.56% | 97.09% |
| Gray&Horizaotal&Crop&Vertical | 92.44% | 91.36% | 94.70% | 91.87% | 92.59% |

Table 14: Detection performance comparison of augmentations.

Existing defense techniques focus either on robust prediction or detection. The most effective way to achieve robust prediction is adversarial training (Elfwing et al., 2018; Zhang et al., 2019), and the use of nearest neighbors is a common approach to detecting AEs. kNN (Dubey et al., 2019) and DkNN (Papernot & McDaniel, 2018) discriminate AEs by checking the label consistency of each layer's neighborhoods. (Ma et al., 2018) define Local Intrinsic Dimensionality (LID) to characterize the properties of AEs and use a simple k-NN classifier to detect AEs. LNG (Abusnaina et al., 2021) searches for the nearest samples in the reference data and constructs a graph, further training a specialized GNN to detect AEs. Although these nearest-neighbor-based methods achieve competitive detection performance, all rely on external AEs for training detectors or searching thresholds, resulting in defeat against unseen attacks.

Recent studies have shown that SSL can improve adversarial robustness as SSL models are label-independent and insensitive to transformations (Hendrycks et al., 2019). An intuitive idea is to combine adversarial training and SSL (Ho & Nvasconcelos, 2020; Kim et al., 2020; Moayeri & Feizi, 2021), which remain computationally expensive and not robust to adaptive attacks. (Shi et al., 2021) and (Mao et al., 2021) find that the auxiliary SSL task can be used to purify AEs, which are shown to be robust to adaptive attacks. However, (Croce et al., 2022) shows these adaptive test-time defenses can be broken by stronger adaptive attacks.

