# OpenReview forum: "Be Your Own Neighborhood: Detecting Adversarial Example by the Neighborhood Relations Built on Self-Supervised Learning"
_ICLR.cc/2024/Conference — Submitted to ICLR 2024_

### Official Review · Reviewer_J3f8 · 2023-10-21

**Soundness:** 3 good
**Presentation:** 3 good
**Contribution:** 3 good
**Rating:** 8
**Confidence:** 4

**Summary:**

The paper tackles the problem of defenses against adversarial examples (AE) targeting deep neural networks (DNN). The main contribution is "BEYOND", i.e., a novel countermeasure which seeks to _detect_ AE by leveraging self-supervised learning (SSL) techniques. The intuition behind BEYOND is that, given a sample, it is possible to create "augmented" versions of such a sample, thereby creating the "neihborhood" of a given sample: then, by using SSL, it is possible to predict the ground truth of the original sample and its neighbors: if the class is the same, then the sample is clean; however, if the class is different, then the sample is an AE. Despite leveraging a simple intuition, the proposed BEYOND is theoretically grounded, and empirically evaluations demonstrate its effectiveness.

All in all, I believe this paper to be a valuable contribution to ICLR.

**Strengths:**

+ Well written and easy to follow
+ The intuition is simple, but sensible
+ The code is disclosed (and it is well written!)
+ The method is theoretically grounded
+ The evaluation is comprehensive (both from the "attacker" perspective, as well as from the considered defenses)
+ An ablation study is carried out
+ Considerations on the implementation costs are provided

I thank the authors for carrying out this work and submitting this paper to ICLR! I really enjoyed reading it. Specifically, I commend them for being able to condense all the "relevant" parts of their contribution into a 9-pages long paper. I could not find any redudancy in the arguments, and the underlying intuition, theoretical explanations, and extensive evaluation clearly demonstrate that the proposed method is valuable. What I particularly appreciated, however, was the discussion/analysis of the implementation costs, wherein the authors acknowledge that the proposed method may have some computational overhead (as a "tradeoff").

It is my belief the work described in this paper has abundant scientific merit.

**Weaknesses:**

## High Level

- The method appears to be limited to the Computer Vision domain
- Some unclear details in the evaluation

The first weakness is what prevents me from assigning "only" an 8 (instead of a 10 -- albeit my score is more leaning towards a 7 than an 8). The entire paper is tailored for DNN applications for Computer Vision. It would be enticing to see how well the proposed method could be "adapted" to cover other domains in which DNN have found applications (e.g., audio, finance, cybersecurity).

For the second weakness (which is the reason why I am "more leaning towards a 7 than an 8"), I was unable to determine if the results provided in the paper refer to a "single" run of a given method, or are provided after performing multiple runs and then averaging the results (if so, please provide the amount of runs as well as the standard deviation).

## Minor comments and suggestions

In the Introduction, the text states the following:

> This vulnerability prevents DNN
from being deployed in safety-critical applications such as autonomous driving Cococcioni et al.
(2020) and disease diagnosis Kaissis et al. (2020), where incorrect predictions can lead to catastrophic
economic and even loss of life.

Tone this down. Aside from it being incorrect (i.e., DNN _are_ deployed in those applications, e.g., [https://spectrum.ieee.org/self-driving-cars-2662494269]), it is an unnecessary overstatement, and there is no need to mention that incorrect predictions can lead to ```loss of life``` -- especially since there is little evidence that such catastrophic events are due to incorrect predictions stemming from "adversarial examples" (and not just due to misconfigurations of the overall system)

Also, I invite the authors to refrain from using "white-/gray-box" terminology to denote the envisioned attacker. According to some recent works from IEEE SaTML23, these terms are ambiguous. I encourage the authors to use "perfect/limited knowledge" attackers. Plus, I also invite the authors to provide a section (in the Appendix) which clearly defines the knowledge/capabilities of the attacker w.r.t. the attacked system. Such a threat model can be justified with a practical example which elucidates a practical use case.

Finally, the reference by "Dan Hendrycks" appears twice in the bibliography.

**Questions:**

While I appreciated the paper, I am willing to increase my score if the authors provide compelling evidence that:

Q1) the method can be applied to different domains (potentially substantiated with some proof-of-concept experiment)

Q2) the results encompass various repetitions (which have been averaged)

---

> ### Author Response · Authors · 2023-11-20
>
> We appreciate your detailed and constructive comments, and we are encouraged that you find our work “well written, easy to follow”, and “valuable contribution”. We explain your questions as follows:
>
> ## Q1 Different Domain
> We understand the reviewer is interested in exploring the use of BEYOND in new domains. We try to migrate our approach to different domains like Natural Language Processing (NLP) and Speech Recognition (SR). But we note some critical differences: existing self-supervised models in NLP or SR are not based on diversified data augmentation strategies. For instance, in NLP, masked token prediction is the primary method. However, having diverse augmentation types is important for BEYOND, because it needs good data augmentation to create neighboring data. Nonetheless, we believe that our approach pioneers an idea of detection based on testing time-augmented examples rather than reference examples. And we will also continue to explore ways to apply this idea to different domains.
>
> ## Q2 Repeated Experiment
> In this paper, our experimental results come from a single run. To further verify the effectiveness of BEYOND, We repeat the experiments in Tables 1 and 2 of the main paper five times. On Cifar10, the average detection performance for the five attacks is 99.16% with a standard deviation of 0.0117. On ImageNet, the average performance against three attacks is 96.44% with a standard deviation of 0.0426.
>
> ## Minor Comments
> Apologies for the uncritical description, we will correct it to“ where incorrect predictions can lead to unpredictable losses”.
> And about the “white-/gray-box” terminology, we appreciate your experienced suggestions. According to your advice, we refer to the latest work **[RR1]** from IEEE SaTML23, and change the terminology to “limited/perfect knowledge” attack in the the latest pdf.
>
> **[RR1]** Apruzzese G, Anderson H S, Dambra S, et al. “Real Attackers Don't Compute Gradients”: Bridging the Gap Between Adversarial ML Research and Practice. 2023 IEEE Conference on Secure and Trustworthy Machine Learning (SaTML). IEEE, 2023: 339-364.
>
>
> Our responses to all questions have been updated to the latest pdf and highlighted in blue.

---

> > ### Comment · Reviewer_J3f8 · 2023-11-20
> > **Ack**
> >
> > Dear authors, thanks for responding to my review.
> >
> > However, it is still unclear to me if the "latest pdf" already integrates the answers to Q1 and Q2. Specifically, for Q1, would the authors be able to do these experiments before the end of the discussion phase? Whereas, for Q2, I'd like to see an updated version of Table 1 and 2 wherein the cells report the average results (and standard deviation) over the five runs.
> >
> > A way to do so is by reporting, e.g., $99.16 {\scriptsize \pm 0.0117}$ (code: ```$99.16 {\scriptsize \pm 0.0117}$```)
> >
> > Also, there is a typo in the reporting in the paper of [RR1].

---

> > > ### Author Response · Authors · 2023-11-21
> > >
> > > Dear reviewer,
> > >
> > > Thanks for your response.
> > > For Q1, we are working on it and will try our best to provide a response before deadline.
> > > For Q2, we have updated tables 1 and 2 with the mean and standard deviation of 5 runs. Please check it in the latest pdf. Note that LNG is not open-sourced and corresponding results come from the paper. So, we cannot calculate the mean and standard deviation for multiple runs.

---

> ### Author Response · Authors · 2023-11-22
>
> Dear reviewer,
>
> For the possibility of migrating BEYOND to other domains, within the limited time, we did a simple experiment on adversarial text detection for NLP. Using the text-attack tool  **[RR1]**, we generated 1000 adversarial texts on a BERT model trained on the MR sentiment classification dataset using the TextFooler attack **[RR2]**. We then used the chatgpt_paraphraser tool to generate ten paraphrases for each text as augmentations. Finally, we compare the embedding similarity between the samples and their augmentations on BERT. The AUC of the detection is 73.63%. Although there is certainly room for performance improvement, we believe that this pilot study confirms the possibility that BEYOND can be migrated to other domains.
>
> **[RR1]** Morris J X, Lifland E, Yoo J Y, et al. Textattack: A framework for adversarial attacks, data augmentation, and adversarial training in nlp[J]. arXiv preprint arXiv:2005.05909, 2020.
>
> **[RR2]** Jin D, Jin Z, Zhou J T, et al. Is BERT really robust? A strong baseline for natural language attack on text classification and entailment[C]//Proceedings of the AAAI conference on artificial intelligence. 2020, 34(05): 8018-8025.

---

> > ### Comment · Reviewer_J3f8 · 2023-11-22
> > **Ack**
> >
> > Dear authors,
> >
> > thank you for performing this additional proof-of-concept experiment. I will take it into account when discussing the paper with the other reviewers.

---

> > > ### Author Response · Authors · 2023-11-23
> > >
> > > Dear reviewer,
> > >
> > > Thank you immensely for your reply and your recognition of our work!

---

### Official Review · Reviewer_qcuA · 2023-10-29

**Soundness:** 3 good
**Presentation:** 3 good
**Contribution:** 2 fair
**Rating:** 5
**Confidence:** 4

**Summary:**

In this study, a simple yet effective adversarial detection method is proposed. This method applies label consistency check and representation similarity check in the embedding space derived by performing contrastive learning over training data samples and their augmented samples. The experimental study involves state-of-the-art adversarial attack and adversarial sample detection methods. Furthermore, it considers both gray and white-box attack scenarios. The results confirm the superior performance of the proposed detection method over the other adversarial sample detection algorithms.

**Strengths:**

1/ The algorithmic design is simple, yet delivering superior adversarial detection performances. It is interesting to use the contrastive learning technique to enlarge the difference between clean samples and adversarially crafted samples. More specifically, the distance between a clean sample and its augmented samples is much smaller than an adversarially perturbed sample and the corresponding augmented variants in the embedding space derived by contrastive learning. This is the first core contribution in this study.

2/ The second core contribution is to make use of the label consistency step to boost the detection accuracy given varying adversarial attack budge levels.

3/ The experiments offer a comprehensive coverage over different attack settings, attack approaches and adversarial sample detection algorithms.

**Weaknesses:**

1/ Theoretical study is not rigorous. The whole study is based on the assumption that data augmentations can effectively weaken adversarial perturbation. However, this is only an empirical observation, yet without any theoretical justification. It is not convincing to set up further deduction based on this hypothesis.

2/ The proposed detection algorithm requires to set many threshold values. In Algorithm.1, three threshold $T_{cos}$, $T_{label}$ and $T_{rep}$ are applied.  The choice of these threshold values are dataset dependent.  This makes the proposed method difficult to be generalized across different datasets / learning scenarios.

**Questions:**

It would be useful to discuss the sensitivity of the thresholds' values over the detection results.

---

> ### Author Response · Authors · 2023-11-20
>
> We express sincere gratitude for your valuable feedback and constructive comments.
>
> ## Q1 Theoretical study
>
> First, our analyses in the main paper are for the case where augmentation can weaken adversarial perturbations, and it shows that if augmentation can weaken adversarial perturbations, it can be used for adversarial sample detection on the SSL representation space.
>
> We note that it is difficult to conduct an overall theoretical analysis to verify that data augmentation as a whole can always effectively weaken adversarial perturbations because there are many data augmentations and they have different effects on adversarial perturbations. We address your question as to how to determine whether one given augmentation type weakens the adversarial perturbation or not.  Therefore, we analyze which data augmentation can weaken adversarial perturbations, i.e., what conditions must be met for augmentations to be effective.
>
> Effective data augmentation makes the augmented label $y_{aug}$ tend to the ground-truth label $y_{true}$ and away from the adversarial label $y_{adv}$ as follow:
>
> $ ||y_{aug}-y_{true}||_2 $
>
> $ \leq ||y_{aug}-y_{adv}||_2 $
>
> $ \leq ||y_{adv}-y_{true}||_2 $
>
> Since $y_{true}$ is the one-hot encoding, the range of $ ||y_{adv}-y_{true}||_2 $ is $(\sqrt{2}/2, \sqrt{2})$.
>
> The distance is $\sqrt{2}$ when the item corresponding to $y_{adv}$ is 1 in the logits of $y_{adv}$, and $\sqrt{2}/2$ when the item corresponding to $y_{adv}$ and $y_{true}$ both occupy 1/2. Given a SSL-based classifier, $C$, we have:
>
> $
> C(W(x+\delta)) = C(Wx)+\nabla C(Wx)W\delta = y_{true} + \nabla C(Wx)W\delta = y_{aug}
> $
>
> Therefore, the distance between $y_{aug}$ and $y_{true}$ is:
>
> $ ||y_{aug}-y_{true}||_2 = ||\nabla C(Wx)W\delta||_2 $
> $ \leq ||\nabla C(Wx)W||_2 ||\delta||_2 $
>
> $ \leq ||\nabla C(Wx)W||_{2} \epsilon $
>
> where $ ||\delta||_2 $ is bounded by $ \epsilon $ according to the definition of adversarial examples.
>
> To make sure $ ||y_{aug}-y_{true}||_2$
>
> $ \leq ||y_{aug}-y_{adv}||_2 $, then:
>
> $ ||\nabla C(Wx)W||_2 \epsilon \leq \frac{\sqrt{2}}{2} \Rightarrow  ||\nabla C(Wx)W||_2 \leq \frac{\sqrt{2}}{{2\epsilon}} $
>
> In summary, an augmentation can mitigate adversarial perturbation when it satisfies $||\nabla C(Wx)W||_2 \leq \frac{\sqrt{2}}{{2\epsilon}}$.
>
> To further validate our analysis, we generate 1000 adversarial examples by PGD with $\epsilon=8/255$ on Cifar10. The following table shows the ratios for different data augmentations meeting the threshold $\frac{\sqrt{2}}{{2\epsilon}}$.
>
> | Aug | Ratio | Aug | Ratio |
> |:---:|:---:|:---:|:---:|
> | Rotation | 99.9% | Vertical | 25.9% |
> | Crop | 40.7% | Color Jitter | 99.0% |
> | Resize | 74.0% | Gray | 40.6% |
> | Horizontal | 25.9% | Compose | 99.7% |
>
> A higher ratio means the augmentation is more effective. Meanwhile, we perform t-sne visualizations of the representations of clean examples and AEs processed by different augmentation methods in Fig. 10 in the latest pdf. As seen in the figure, the effective augmentation methods with the high ratio in the above table can effectively increase the distance between AEs and their neighbors. For example, Rotation has the highest ratio in the above table, and the distance between AE and its neighbors in the figure is larger than that of clean samples. While Horizontal and Vertical have the lowest ratio, and the distance between AE and its neighbors is still close in the above figure.
>
> Moreover, we test the detection performance of high-ratio augmentations and low-ratio augmentations as follows:
>
> |Augmentation|FGSM|PGD|CW|AutoAttack|Average|
> |:---:|:---:|:---:|:---:|:---:|:---:|
> |ColorJitter&Resize&Rotation| 97.11% | 96.55% | 98.15% | 96.56% | 97.09% |
> |Gray&Horizaotal&Crop&Vertical| 92.44% | 91.36% | 94.70% | 91.87% | 92.59% |
>
> It can be seen that the average detection performance of the effective augmentations obtained by our analysis is 5% higher than that of the other augmentations.
>
> Finally, BEYOND adopts various augmentations instead of a single type to generate multiple neighbors for AE detection. This allows the detection performance to circumvent the effects of bad augmentation methods, which reduces randomness and makes the estimates more robust.
>
> ## Q2 Threshold
> The evaluation metric in our paper is mainly area-under-curve (AUC). BEYOND includes three thresholds. For any threshold set, we compute a True Positive Rate (TPR) and a False Positive Rate (FPR). High AUC on different datasets shows that: 1) BEYOND can get a high TPR when FPR is low; 2) when we choose a bad threshold set that has a high FPR, we also get a high TPR. 3) The sensitivity of BEYOND to the threshold is weaker than baselines. Hence, BEYOND is not threshold-sensitive.
> Moreover, in practice, we can choose a threshold with FPR5% only based on clean examples, which is not allowed for baselines that require adversarial examples as a reference set or training.
>
> Our responses to all questions have been updated to the latest pdf and highlighted in blue.

---

> > ### Comment · Reviewer_qcuA · 2023-11-22
> > **Thanks for the resposne**
> >
> > Dear authors,
> >
> > I appreciate your efforts in addressing my question and I would apologise for my late reply.
> >
> > I have gone through the rebuttal texts. I feel the theoretical study you offered over the condition of the augmentation still needs further investigation.
> >
> > In the inequality \epsilon \leq \frac{2}{2\|\nabla{C}(Wx)W\|_{2}}, it indicates that a larger gradient magnitude of C at Wx requires to reduce the magnitude of the injected augmentation.
> >
> > Let's assume W is well normalized, so we can ignore the effects imposed by \|W\|, the inequality seems to show that it is more difficult to differentiate the impact of injecting an augmentation noise from that caused by adversarial manipulation in highly insensitive area in the input space, i.e. the area with large \|\nabla{C}(Wx)\|. For example, in the extreme case, the augmentation noise that we can impose without changing the label will become marginally small if \|\nabla{C}(Wx)\| becomes very large. In such a case, both adversarial noise and data transformation-based augmentation may cause the drastic output change. Can we say in this scenario that the idea of adding data augmentation to detect adversarial noise fails ?
> >
> > All in all, I think the success of this detection-by-augmentation strategy may also be determined by the sensitivity / curvature of the input area where we apply this detection method.
> >
> > Regarding the threshold question, I understand the threshold values can be chosen by checking the AUC curves. This is the standard protocol that we would also follow in our study. However, my concern is over the situation that this detection algorithm needs three threshold $T_{cos}$, $T_{label}$ and $T_{rep}$, tuning all of them makes the deployment of this work in practices difficulty, more freedom of adjustion, more challenging to maintain / debug the system.
> >
> > Again, I thank the authors' efforts in shaping this study -- it is impressive. But from my perspective, it still needs some improvement to make this study more solid for the top venue like ICLR.

---

> > > ### Author Response · Authors · 2023-11-22
> > >
> > > Dear reviewer,
> > >
> > > Thanks for your responses.
> > >
> > > For Q1, we apologize for not describing the notations clearly, causing your misunderstanding. In the theory analysis, we simplify the augmentation process as a matrix $W$, and we do a First Order Taylor Expansion at perturbation is zero ($\delta = 0$).
> > >
> > > We agree that there will be some extreme augmentations that may lead to drastic output change. But we think that this type of augmentation is going to be carefully designed, possibly based on gradients. However, in our analyses and tests, we are using some common augmentations.
> > >
> > > In addition, we also think that the success of BEYOND may depend on the sensitivity of the input region to the augmentation. However, since we have no prior knowledge of the sensitivity of input regions to certain types of augmentations,  we cannot use only one type of augmentation for detection. We believe that the use of multiple augmentations in BEYOND can cover the so-called sensitive regions and thus enable effective detection of adversarial examples.
> > >
> > > For Q2, your concerns about practical deployment are worthy of our consideration and we will address them in future work. Thank you very much for your valuable suggestions.

---

> ### Author Response · Authors · 2023-11-23
>
> Dear reviewer,
>
> We deeply appreciate your valuable time and the comprehensive feedback you have provided. However, upon reviewing your response, we have identified some misunderstandings that we would like to address and clarify in relation to our work:
>
> 1. The assumption we made in our theoretical analysis regarding the "benign perturbation, i.e. random noise, $\hat{\delta}$, with bounded budgets causing minor variation, $\hat{\epsilon}$, on the feature space" is intuitive and aligns with prior literature **[RR1][RR2][RR3]**. We believe this assumption is reasonable and aligns with established research in the field.
>
> 2. As for your concern about ''$|\nabla{C}(Wx)|$ becoming very large'', we believe that such a scenario is not realistic. This is because the SSL-based classifier $C(\cdot)$ that we employ has already undergone parameter optimization using \textbf{gradient descent} during the training process. Therefore, in situations where $C(\cdot)$ can achieve high accuracy, it is unlikely to encounter a situation where $|\nabla{C}(Wx)|$ becomes large.
>
> 3. Furthermore, it is important to note that the utilization of the SSL model offers the advantage of producing stable features even under common augmentations. This characteristic helps to mitigate the occurrence of extreme cases mentioned earlier to a significant extent. The use of SSL models allows for more robust and reliable performance, thereby enhancing the overall stability of the system.
>
> We apologize for any misunderstandings that may have arisen.
> Finally Thank you again for the truly splendid discussion, and if you have any further concerns or questions, please do not hesitate to let us know.
>
> **[RR1]** Nesti F, Biondi A, Buttazzo G. Detecting adversarial examples by input transformations, defense perturbations, and voting[J]. IEEE Transactions on neural networks and learning systems, 2021.
>
> **[RR2]** Tian S, Yang G, Cai Y. Detecting adversarial examples through image transformation[C]//Proceedings of the AAAI Conference on Artificial Intelligence. 2018, 32(1).
>
> **[RR3]** Xu W, Evans D, Qi Y. Feature squeezing: Detecting adversarial examples in deep neural networks[J]. arXiv preprint arXiv:1704.01155, 2017.

---

> > ### Comment · Reviewer_qcuA · 2023-11-23
> > **Thanks for the follow up discussion**
> >
> > However, even for a converged classifier, there are still high curvature areas, for which the gradient magnitudes regarding to individual data points remaining high. Usually they correspond to data points difficult to fit by the model (can be rare-class samples or containing natural noise inside).
> >
> > Besides, there are also vulnerable data points where a slight change to the data points would cause large fluctuation to the decision output. We can use influence score [LiangICML2017] to identify such data points.
> >
> > LiangICML2017. Liang et al, Understanding Black-box Predictions via Influence Functions, ICML 2017.
> >
> > Even for SSL, these vulnerable data points exist. Of course, these cases are usually in the long-tailed area of data distribution. However, in these cases, using data augmentation to detect the difference between augmented data points and adversarially perturbation can fail to give a correct answer. The related discussion is absent from the current discussion. I don't see how the current algorithmic design can offer a guarantee to which extent it can handle / avoid such a situation.
> >
> > Again, I am not denying the effectiveness of your approach. My concern is over the completeness of the analysis over your approach presented in this paper. If any practitioner uses this approach for their applications, it is important for them to understand the feasibility condition of the proposed method and when/how this method may fail.

---

### Official Review · Reviewer_cka9 · 2023-10-31

**Soundness:** 3 good
**Presentation:** 2 fair
**Contribution:** 2 fair
**Rating:** 6
**Confidence:** 3

**Summary:**

This paper proposes an adversarial detection method called BEYOND, which detects the adversarial examples using label consistency and representation similarity with neighbors.

**Strengths:**

1. The mathematical analysis is logical and convincing when combining with the proposed detection structure.

2. The conflicting goals for adaptive attacks against the proposed method is original.

**Weaknesses:**

1. The baselines selected in the paper are somewhat old. Using baselines with the same properties such as neighbors and representations is reasonable, while we believe that comparisons with newer methods with or without such properties are necessary, such as SimCLR for catching and categorizing (SimCat) [1] which also use representations effectively and Erase-and-Restore (E&R) [2].

2. The format of citations is incorrect. For example, "kNN Dubey et al. (2019)" in Baselines of Section 4 should be "kNN (Dubey et al., 2019)".

3. The detection ability for various types of attacks is beneficial for its applications, thus I am concerned about the evaluations of detection the adversarial samples generated by attacks based on different norm.

[1]Moayeri M, Feizi S. Sample efficient detection and classification of adversarial attacks via self-supervised embeddings[C]//Proceedings of the IEEE/CVF international conference on computer vision. 2021: 7677-7686.

[2]Zuo F, Zeng Q. Exploiting the sensitivity of L2 adversarial examples to erase-and-restore[C]//Proceedings of the 2021 ACM Asia Conference on Computer and Communications Security. 2021: 40-51.

**Questions:**

Please see the Weaknesses section.

==============After rebuttal===============
The explanations and results provided by authors address most of my concerns. Thus, I am willing to raise the rating score.

---

> ### Author Response · Authors · 2023-11-20
>
> We express sincere gratitude to your valuable feedback and constructive comments.
>
> ## Q1 Baselines
> Our proposed BEYOND focuses on detecting adversarial examples by leveraging the relationship between the input and its neighbors. So, the baseline methods we chose are all related to detectors that are built on the distance between the input sample and its neighbors. And since our method uses the SSL model, we also compare BEYOND with Mao et al, which is a SSL-based adaptive-robust defense method.
>
> Moreover, we also included the suggested baselines SimCat from “**[RR1]**” and  E&R from  “**[RR2]**”. The comparison on ImageNet for CW L2 attack is as follows:
> | AUC(%) | BEYOND | SimCat | E&R |
> |:---:|:---:|:---:|:---:|
> | C&W L2 | **96.32%** | 81.85 | 93.57 |
>
> Please note that SimCat is not open source, so the corresponding results come from our own replication. Erase-and-Restore (E&R) is only for L2 norm attacks.
> It can be seen that BEYOND outperforms these two baselines. In addition, both SimCat and E&R require training a detector, which is not required inBEYOND.
> For adaptive attacks, Table 5 in **[RR1]** shows that as a few-shot detector, SimCat itself is not robust to adaptive attacks, and its adaptive robustness can only be improved by combining a robust model (e.g. Adversarial training model).
> E&R in **[RR2]** evaluated its robustness to adaptive attack (BPDA), but **[RR3]** shows BPDA is not as effective as APGD used in BEYOND.
>
> **[RR1]** Moayeri M, Feizi S. Sample efficient detection and classification of adversarial attacks via self-supervised embeddings[C]//Proceedings of the IEEE/CVF international conference on computer vision. 2021: 7677-7686.
>
> **[RR2]** Zuo F, Zeng Q. Exploiting the sensitivity of L2 adversarial examples to erase-and-restore[C]//Proceedings of the 2021 ACM Asia Conference on Computer and Communications Security. 2021: 40-51.
>
> **[RR3]** Francesco Croce, Sven Gowal, Thomas Brunner, Evan Shelhamer, Matthias Hein, and Taylan Cemgil. Evaluating the adversarial robustness of adaptive test-time defenses. In International Conference on Machine Learning, pp. 4421–4435. PMLR, 2022
>
> ## Q2 Citation Format
> Thanks for your careful check, we have corrected the citation format in the latest pdf.
>
> ## Q3 Various Types of Attacks
> Your concern about robustness to various types of attacks is warranted. Following the reviewer’s comments we test the most representative method that supports multiple norm attacks, AutoAttack. AutoAttack supports $L_{\infty}$, $L_2$ and $L_1$ norm attacks. In the main paper, we only report the detection performance of BEYOND against AutoAttack $L_{\infty}$. The following table shows the performance of BEYOND against AutoAttack with different norms.
>
> | AUC(%) | $L_{\infty}$ | $L_2$ | $L_1$ |
> |:---:|:---:|:---:|:---:|
> | Cifar10 | 99.18 | 99.13 | 99.07 |
> | ImageNet | 97.14 | 97.26 | 97.18 |
>
> The perturbation budgets($\epsilon$) on Cifar10 are 8/255 ($L_{\infty}$), 0.5 ($L_2$), and 8 ($L_1$); and on ImageNet are 8/255 ($L_{\infty}$), 3 ($L_2$), and 64 ($L_1$). The result shows BEYOND is still effective against attacks based on different norms.
>
> Our responses to all questions have been updated to the latest pdf and highlighted in blue.

---

> > ### Comment · Reviewer_cka9 · 2023-11-22
> > **Response to authos**
> >
> > Dear authors
> >
> > Thanks a lot for your careful responses. I believe the provided explanations and results address most of my concerns. Thus, I will raise the rating score.

---

> > > ### Author Response · Authors · 2023-11-22
> > >
> > > We thank the reviewer for the endorsement! Your comments and suggestions are instrumental to our work.

---

### Official Review · Reviewer_SDmS · 2023-11-01

**Soundness:** 3 good
**Presentation:** 2 fair
**Contribution:** 3 good
**Rating:** 6
**Confidence:** 3

**Summary:**

The paper proposes BEYOND, an adversarial example (AE) detection method which is based on label and representation consistency of augmented neighbor samples using a pretrained SSL model.

The method builds on ideas from DkNN [A] and LNG [B].

[A] Nicolas Papernot and Patrick McDaniel. Deep k-nearest neighbors: Towards confident, interpretable and robust deep learning. arXiv preprint arXiv:1803.04765, 2018.

[B] Ahmed Abusnaina, Yuhang Wu, Sunpreet Arora, Yizhen Wang, Fei Wang, Hao Yang, and David Mohaisen. Adversarial example detection using latent neighborhood graph. In Proceedings ofthe IEEE/CVF International Conference on Computer Vision, pp. 7687–7696, 2021.

The paper claims that the above-mentioned AE detection methods have limitations. Some AEs required to build the graph, and they cannot generalize to unseen attacks. They can be bypassed by adaptive attacks.

There's a theoretical analysis provided in the paper which explains the reasoning behind the applicability of the core idea of the paper. The conclusion of the analysis is that the imperceptible perturbation δ in the image space can be significantly enlarged in SSL’s feature space, and this can be detected by referring to the original image's neighbors.

The proposed method can be used with Adversarially Trained models and is robust to adaptive attacks. The paper claims that the robustness to adaptive attacks comes from the conflicting optimization goals for the attacker where there is a cancelation of gradients, leading to poor adaptive attacks.

**Strengths:**

The AE detection method proposed in the paper uses a novel SSL model-based approach. The experiments are thorough to support the claims. The method proposed is robust to adaptive attacks.

**Weaknesses:**

The writing for the experiments section can be substantially improved. The main conclusions from the analysis can be highlighted better in text by shortening details. The figure captions should be made self-contained. It's hard to parse the figures independently of the text.

**Questions:**

1. (Section 2.1) The paper claims, “Note that BEYOND is not based on random data augmentation.” But in Section 4.1, the paper says, “Augmentations BEYOND uses for generating neighbors are consistent with SimSiam, including horizontal flips, cropping, color jitter, and grayscale.” Aren't SimSiam augmentations random? A clarification will be helpful.

2. (Typo) Section 4.1 “a more IMAGENET”

3. (Repeated citation names) “Hu Hu et al. (2019)” and "Mao Mao et al. (2021).

---

> ### Author Response · Authors · 2023-11-20
>
> ## Weakness:
> Thanks for your useful comments, we will adjust the paper to enhance readability. For the figure caption, we have added the necessary details to make it self-contained in the final paper.
>
> ## Q1 Augmentation Type
> We apologize for the confusion. The type of augmentations used by BEYOND is consistent with SimSiam's, including horizontal flips, cropping, color jitter, and grayscale. Since we need to generate multiple different neighbor samples for each input image, we use different combinations of augmentation parameters. However, unlike SimSiam's random augmentations, we fix the random seed to ensure that our method does not benefit from randomization.
> We have modified the corresponding part to make the method setting clearer.
>
> ## Q2 & Q3 Typo and Citation
> Sorry for the discomfort caused by typos and repeated citation names, we have carefully proofread our paper.
>
> Our responses to all questions have been updated to the latest pdf and highlighted in blue.

---

### Author Response · Authors · 2023-11-23

Dear Reviewers and ACs,

Thank you all for your time and effort in reviewing this paper. We are grateful for the positive recognition by all the reviewers.

We summarize our paper's main contributions, including the additional conclusions during the rebuttal discussion phase:

* We proposed BEYOND, a novel SSL-based AE detection framework, which takes advantage of the neighborhood relations built on SSL models. To our knowledge, BEYOND is the first work that leverages an SSL model for AE detection without prior knowledge of adversarial attacks or AEs.

* We develop a rigorous justification for the effectiveness of BEYOND, and we derive an indicator to evaluate the validity of the candidate augmentation.

* BEYOND's performance against standard adversarial attacks and adaptive attacks is better than SOTA。

* BEYOND pioneers an idea of detection based on testing time-augmented examples rather than reference examples, and preliminary experiment shows that it has the potential for application in different domains.

We thank all the reviewers again for actively engaging in the rebuttal discussion and for their positive recognition of our work.

Sincerely, Paper 7036 Authors.

---

### Meta-Review · Area_Chair_K2C2 · 2023-12-23

**Metareview:**

Thank you for submitting to ICLR 2024!

Beyond is an adversarial example detection method that relies on label consistency and representation similarity in a self-supervised representation to score inputs as adversarial or not. The consistency and similarity are measured across each input and "neighbors" made by data augmentation from the input. If either the label consistency or representation similarity do not meet their respective thresholds then the input is rejected as adversarial. Like other test-time defenses, incl. Mao, Beyond does not require adversarial examples during training but it can be combined with adversarial training. Compared against other detection-based defenses, Beyond achieves the best adversarial and natural accuracies (Tab. 1 & 2 on CIFAR-10 and ImageNet resp.) with a total inference time that is only ~2x that of the base classifier and faster than other defenses like Mao or LNG (Tab. 5). An adaptive attack is presented with the use of EoT and theoretical analysis is provided in line with prior work.

Four expert reviewers are divided and rate the submission as accept (J3f8), borderline accept (SDmS, cka9), and borderline reject (qcuA). Reviewer cka9 raised their score due to the author response. During the reviewer-AC discussion phase, Reviewer J3f8 championed the submission due to its positive empirical results while Reviewer qcuA argued against it due to insufficient theoretical justification and overlap with prior work (in particular: [Lecuyer et al. Oakland 2019](https://arxiv.org/abs/1802.03471), [Li et al. NeurIPS 2019](https://arxiv.org/abs/1809.03113), and [Cohen et al. ICML 2019](https://arxiv.org/abs/1902.02918) in addition to the already cited work of Mao et al.). Lastly, Reviewer cka9 concurs with qcuA about the need for more clarity about. the novelty and theoretical justification of Beyond w.r.t. these existing works, although cka9 is more positive.

The AC sides with rejection, given the borderline scores, but underlines that there is merit in this work and encourages the authors to revise it and resubmit. The issues are that the experimental and theoretical content are not fully situated and justified w.r.t. prior work and in line with reviewer expectations. This is evidenced by the diverging borderline reject score and the strong positions taken for and against during reviewer discussion of the submission. Please see the "Why Not Higher" section for further detail on how the submission can be improved for future submission to a venue such as ICML 2024 or NeurIPS 2024.

Note: The AC thanks the authors for the specific and polite confidential comment, and acknowledges the highlighting of the use of self-supervised learning and the selection of neighbors by augmentation of input samples vs. reference samples.

Note: The authors are advised to revise the references to cite the published editions of works, where published, rather than the arXiv editions (this was not a factor in the decision, but is merely a friendly point of writing feedback).

**Justification For Why Not Higher Score:**

- This submission needs to clearly delineate its theoretical justification from the papers on randomized smoothing raised during the reviewer-AC discussion phase (Lecuyer et al., Li et al. and Cohen et al.) and/or cite these works where necessary.
- There is only partial evaluation under the stronger adaptive attacks of Croce et al. ICML 2022 that make use of transfer from the static model (without the defense), BPDA, and EoT together. This Croce et al. analysis paper on adaptive defenses is cited, so it is known, but its attacks applied to related self-supervised defenses such as Mao et al. (ICCV 2021) are not evaluated. Mao is closely related and its analysis and status as a defense is relevant to this submission: Mao et al. has already analyzed self-supervised scores of natural and attacked inputs and their neighbors (see their Figure 2), setting a precedent for Beyond. It is worth noting that Mao et al. has been broken by [Croce et al. ICML'22](https://arxiv.org/abs/2202.13711), and so this may reflect a potential weakness in Beyond as a related method, although of course this conclusion cannot be extended to Beyond without experiment. These experiments should be done to reinforce the evaluation against adaptive attack.
- There is missing related work on adversarial example detection or mititgation by transformations and neighbor relations. See for instance Runtime Masking and Cleansing (ICML 2022), which also makes use of neighbor relations during inference (though it does so to update the model and it must precompute adversarial examples as reference neighbors), and Enhancing Adversarial Robustness via Test-time Transformation Ensembling (ICCVW 2021) which ensembles over input transformations and may explain away some of the robustness boost from the proposed BEYOND. To be clear, there are differences, and in this work the neighbors are entirely augmentations of the input and the representation is self-supervised, but the relations remain and deserve discussion.
- Ablation is needed to understand the relative contribution of label consistency vs. representation similarity and to gauge how much Beyond contributes past simple test-time augmentation of the input and averaging over predictions (so-called test-time ensembling—see Test-time Transformation Ensembling above).

**Justification For Why Not Lower Score:**

N/A

---

### Decision · Program_Chairs · 2024-01-16

Reject